# MVP-Net: Multi-View Depth Image Guided Cross-Modal Distillation Network for Point Cloud Upsampling

Jiade Chen
Beijing University of Technology
Beijing, China
jdchen@emails.edu.cn

Jin Wang*
Beijing University of Technology
Beijing, China
ijinwang@bjut.edu.cn

Yunhui Shi
Beijing University of Technology
Beijing, China
syhzm@bjut.edu.cn

Nam Ling
Santa Clara University
Santa Clara, USA
nling@scu.edu.cn

Baocai Yin
Beijing University of Technology
Beijing, China
ybc@bjut.edu.cn

## Abstract

Point cloud upsampling concerns producing a dense and uniform point set from a sparse and irregular one. Current upsampling methods primarily encounter two challenges: (i) insufficient uni-modal representations of sparse point clouds, and (ii) inaccurate estimation of geometric details in dense point clouds, resulting in suboptimal upsampling results. To tackle these challenges, we propose MVP-Net, a multi-view depth image guided cross-modal detail estimation distillation network for point cloud upsampling, in which the multi-view depth images of point clouds are fully explored to guide upsampling. Firstly, we propose a cross-modal feature extraction module, consisting of two branches designed to extract point features and depth image features separately. This setup aims to produce sufficient cross-modal representations of sparse point clouds. Subsequently, we design a Multi-View Depth Image to Point Feature Fusion (MVP) block to fuse the cross-modal features in a fine-grained and hierarchical manner. The MVP block is incorporated into the feature extraction module. Finally, we introduce a paradigm for multi-view depth image-guided detail estimation and distillation. The teacher network fully utilizes paired multi-view depth images of sparse point clouds and their dense counterparts to formulate multi-hierarchical representations of geometric details, thereby achieving high-fidelity reconstruction. Meanwhile, the student network takes only sparse point clouds and their multi-view depth images as input, and it learns to predict the multi-hierarchical detail representations distilled from the teacher network. Extensive qualitative and quantitative results on both synthetic and real-world datasets demonstrate that our method outperforms state-of-the-art point cloud upsampling methods.

## CCS Concepts

• **Computing methodologies** → **Computer vision**.

---

*Corresponding author.

## Keywords

Point Cloud Upsampling; Deep Neural Networks

**ACM Reference Format:**

Jiade Chen, Jin Wang, Yunhui Shi, Nam Ling, and Baocai Yin. 2024. MVP-Net: Multi-View Depth Image Guided Cross-Modal Distillation Network for Point Cloud Upsampling. In *Proceedings of the 32nd ACM International Conference on Multimedia (MM '24), October 28–November 1, 2024, Melbourne, VIC, Australia.* ACM, New York, NY, USA, 10 pages. https://doi.org/10.1145/3664647.3681562

## 1 Introduction

As a 3D data format, point clouds are widely used in many fields, including autonomous driving [26] and 3D reconstruction [25], thanks to their effective representation capability and the advancements in 3D sensing technology. However, the inherent limitations of 3D sensing devices often result in point clouds being sparse, non-uniform, noisy, and containing outliers. These drawbacks pose a hindrance to the practical application of point clouds. Therefore, converting sparse point clouds into high-fidelity versions that accurately represent the underlying 3D shapes is crucial. To this end, various point cloud upsampling techniques such as [21–23, 28, 37] have been developed.

The predominant methods for point cloud upsampling are those based on deep learning [21, 23, 28, 31, 40]. As illustrated in Figure 1(a), most of these methods consist of two fundamental modules: feature extraction and upsampling tail. Initially, the feature extraction module is employed to extract point features. Subsequently, the upsampling tail is utilized to expand point features and regress the 3D coordinates of upsampled points, thereby achieving upsampling. However, these methods encounter two primary issues. The first issue is that these methods rely solely on the 3D coordinates of sparse point clouds to predict their dense counterparts. They do not fully utilize other available modality information, resulting in insufficient uni-modal representations of sparse point clouds. The second issue is the inaccurate estimation of geometric details in dense point clouds, resulting in a lack of fidelity in the fine geometric details in the upsampled results (refer to Figure 5). Existing methods utilize KNN [3, 27] or attention [34] mechanisms to capture the geometric details of point clouds. However, the former is limited by its receptive field and has restricted modeling capabilities. Meanwhile, the latter faces challenges in capturing geometric details due to the absence of inductive bias regarding

them. To tackle these two issues, we introduce a **M**ulti-**V**iew Depth Image Guided Cross-Modal Distillation Network for **P**oint Cloud Upsampling (**MVP-Net**) as shown in Figure 1(b). With MVP-Net, we explore the optimal utilization of multi-view depth images to effectively guide point cloud upsampling.

Firstly, we argue that alongside the 3D coordinates, feature information from other modalities can also provide complementary information to assist the precise reconstruction of 3D point clouds. The 2D depth image of a 3D point cloud not only contains the depth value of each individual point but also provides information, reflecting the shape, contour, and details of the underlying 3D object. However, capturing such information using only 3D coordinates is usually challenging due to the sparsity, irregularity, non-uniformity, noise, and outliers in the point cloud. Therefore, we propose a cross-modal feature extraction module to comprehensively utilize the cross-modal feature information from both 3D point clouds and 2D depth images. It contains dual branches to extract both point features and depth image features. In the point cloud branch, each point cloud is treated as a graph and graph convolutions are utilized to capture local details. In the depth image branch, the multi-view depth images of the point cloud are taken as input, and 2D convolution is employed. We then introduce a **M**ulti-**V**iew Depth Image to **P**oint Feature Fusion (**MVP**) block, which is integrated into the feature extraction module and effectively fuses these cross-modal features in a fine-grained and hierarchical manner.

Secondly, though the 2D depth image provides complementary information, this information is somewhat limited due to the sparsity of the sparse point cloud. In contrast, the dense point cloud includes rich detail information, as well as its depth image. Therefore, we introduce a detail estimation and distillation structure to make full use of detail information from dense point clouds. Specifically, we establish both a teacher network and a student network. The teacher network uses the multi-view depth images of dense point clouds as additional inputs. Therefore, a well-trained teacher network can effectively and accurately reconstruct fine geometric details and form multi-hierarchical detail representations, thereby demonstrating excellent performance (refer to Tables 1 and 2). Next, the student network is trained using only the sparse point cloud and its multi-view depth images. It learns to predict the multi-hierarchical detail representations from the teacher network. This enables the student network to produce finer geometric details in the upsampled results without relying on extra information. Performance evaluations across various datasets and settings demonstrate that MVP-Net achieves state-of-the-art performance.

In summary, our contributions are as follows:

- We first propose to use the multi-view depth images of a point cloud as guidance to point cloud upsampling, then we present a novel network, MVP-Net, which makes full use of cross-modal representations and thus can accurately estimate geometric details.
- A cross-modal feature extraction module and a Multi-View Depth Image to Point Feature Fusion (MVP) block are designed. The former consists of dual branches to extract both point features and depth image features, while the latter is integrated into the former to fuse these cross-modal features in a fine-grained and hierarchical manner, thereby generating sufficient cross-modal representations.

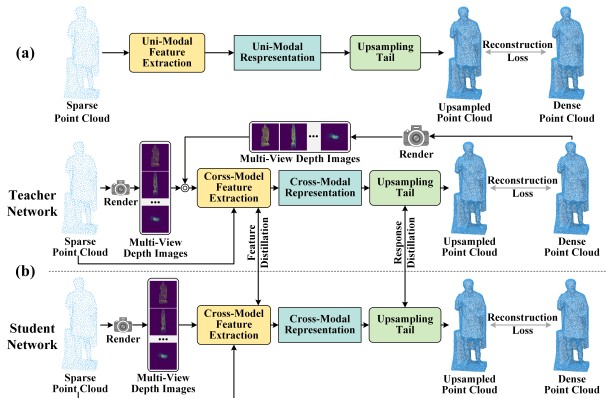

Figure 1: (a) Conventional point cloud upsampling methods mainly depend on 3D coordinates to directly estimate dense point clouds. (b) We introduce a multi-view depth image guided cross-modal detail estimation distillation network for point cloud upsampling, which can generate sufficient cross-modal representation of sparse point clouds. The teacher network captures geometric details in dense point clouds, using paired multi-view depth images. Then the knowledge of geometric details is distilled into the student network.

- A distilling structure is introduced to produce finer geometric details in upsampled point clouds, guided by the multi-view depth image. The rich detail information from the dense point cloud is derived by the teacher network and transferred to the student network, facilitating the generation of geometric details without relying on extra information.
- Extensive experimental results show that our method outperforms state-of-the-art point cloud upsampling methods on both synthetic and real-world datasets.

## 2 Related Works

### 2.1 Point Cloud Upsampling

With the significant success of deep learning technology in point cloud processing, deep learning-based methods have become mainstream in the field of point cloud upsampling [6, 13, 21, 23, 24, 28–30, 35, 40]. PU-Net [40] is a pioneering deep learning-based work, which expands a point set via a multi-branch convolution in the feature space. Then, MPU [37] proposes to progressively upsample point patches in multiple steps. PU-GAN [21] introduces to employ a generative adversarial network for upsampling. PUFA-GAN [23] adopts a frequency-aware discriminator to tackle the issue of noise in the generated shape. PU-GCN [28] uses graph convolutions and shuffle operations to expand features. In addition, PUGeo-Net [29] initially generates points on 2D the tangent plane and then evolves them into 3D space. MAFU [30] incorporates a similar concept to PUGeo-Net to achieve flexible upsampling. Furthermore, Dis-PU [22] leverages a dense generator and a spatial refiner to achieve upsampling by disentangled refinement. PUCRN [6] generates points in multiple steps and then adjusts the spatial positions of each point. PU-Flow [24] employs the normalizing flow model to achieve upsampling by interpolating in latent space. NeuralPoints

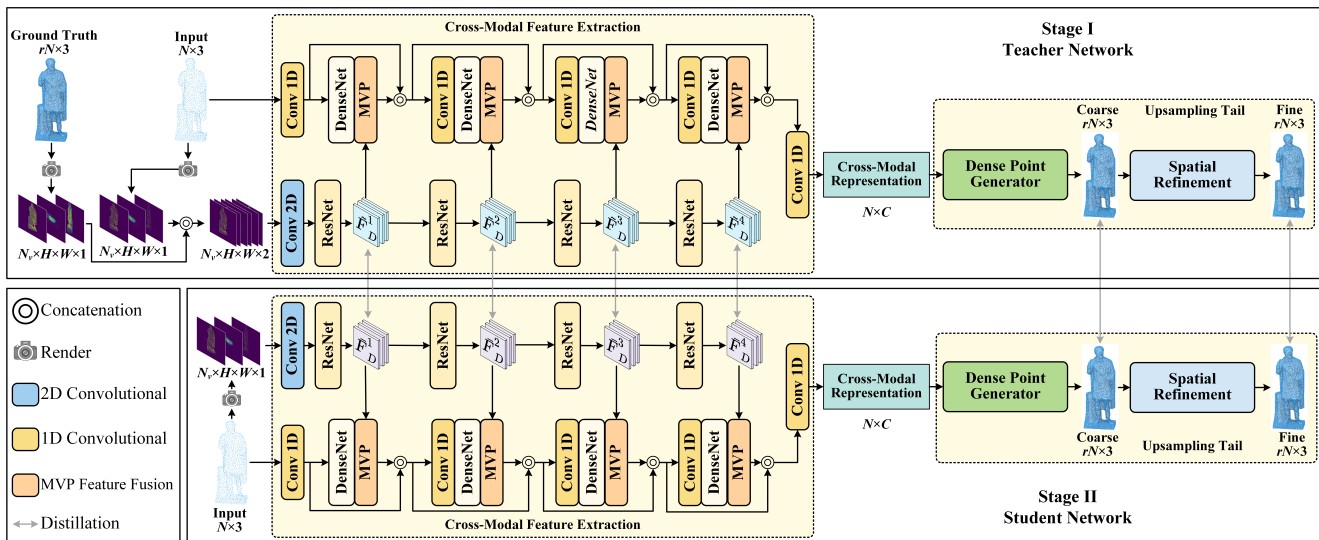

**Figure 2: The overall architecture of MVP-Net. We first train the teacher network (top) with additional multi-view depth images of ground truth. Then, we train the student network (bottom) to learn the knowledge distilled from the teacher network.**

[8] employs the neural implicit functions to represent the local neural fields to achieve arbitrary upsampling. Recently, SSPU-Net [35] introduces a structure-sensitive upsampling network to handle and reconstruct complex geometric structures. Grad-PU [13] utilizes the idea of gradient descent with learned distance functions to achieve arbitrary-scale upsampling. However, these above methods only utilize 3D coordinates to learn the mapping from sparse point clouds to their dense counterparts. To overcome this limitation, we propose MVP-Net, which possesses the capability to generate sufficient cross-modal representations of sparse point clouds and produce finer geometric details in upsampled point clouds.

## 2.2 Cross-Modal Point Cloud Reconstruction

Point cloud completion and point cloud upsampling are both tasks related to point cloud reconstruction. The former involves predicting a complete point cloud from an incomplete one, and some cross-modal methods have been developed for this purpose. For example, ViPC [43] proposes to leverage a pre-trained single-view reconstruction model to infer complete shapes. Subsequently, XMFNet [1] leverages direct cross-modal information fusion at a feature level, avoiding explicit reconstruction. CSDN [46] proposes to transfer the intrinsic shape characteristics from single images to guide the geometry generation of the missing regions. Recently, SVD-Former [45] incorporates multi-view depth images of incomplete point clouds to predict global complete shape by a self-view fusion network. However, these cross-modal completion methods fuse cross-modal features in a coarse-grained manner and focus on leveraging information from other modalities to predict complete shapes, rather than generating geometric details. Therefore these cross-modal methods cannot be directly used for point cloud upsampling. Currently, there is a shortage of cross-modal methods tailored for point cloud upsampling. To this end, we propose a dedicated cross-modal feature extraction module for upsampling, along with a fine-grained cross-modal feature fusion block that integrates pixel-level depth image features into corresponding point features.

## 2.3 Knowledge Distillation for Point Cloud Learning

A common application of knowledge distillation for point cloud learning, similar to its original concept introduced by [14], is for model compression [11, 15, 20, 42]. For the point cloud reconstruction task, RaPD [7] proposes a reconstruction-aware prior distillation method for point cloud completion, where the student network learns the distribution of the latent space generated by the teacher network. Recently, CPU [44], which introduces vector-quantization into upsampling, incorporates knowledge distillation to facilitate effective codebook learning and enhance the utilization of code entries in the student network. We have a different purpose, which is to utilize the teacher network to capture geometric details in dense point clouds and then transfer this knowledge to the student network, guided by multi-view depth images.

## 3 Method

### 3.1 Overview

Point cloud upsampling aims to produce a dense point cloud $Q \in \mathbb{R}^{rN \times 3}$ of $rN$ points from a sparse one $\mathcal{P} \in \mathbb{R}^{N \times 3}$ of $N$ points, where $r$ denotes the upsampling ratio. Given a sparse point cloud $\mathcal{P}$, we employ a basic renderer [10] to generate multi-view depth images $\mathcal{I}_{\mathcal{P}} \in \mathbb{R}^{N_v \times H \times W \times 1}$, where $N_v$ is the number of viewpoints and $H, W$ denote the resolution of each depth image. We first train a teacher network $\mathcal{T}$ using additional multi-view depth images $\mathcal{I}_{\hat{Q}}$ of ground truth $\hat{Q}$ to derive the geometric details. Then, we train a student network $\mathcal{S}$ to learn the knowledge from the teacher network. In the following sections, we first describe the network structure of both the teacher network and the student network, which include a cross-modal feature extraction module (see Sec. 3.2) integrated with the cross-modal feature fusion block (see Sec. 3.3) and an upsampling tail (see Sec. 3.4). Then, we introduce the structure of detail estimation and distillation (see Sec. 3.5). Finally,

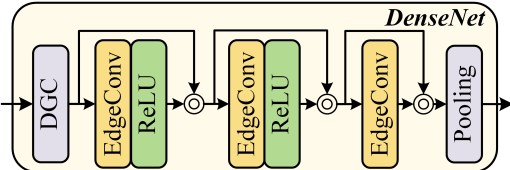

**Figure 3: The architecture of DenseNet in cross-modal feature extraction module. DGC denotes the dynamic graph construction operation [36].**

Sec 3.6 presents the training objectives of the teacher network and the student network.

## 3.2 Cross-Modal Feature Extraction

The cross-modal feature extraction module takes a sparse point cloud $\mathcal{P}$ and its multi-view depth images $I_{\mathcal{P}}$ as input. To fully extract the features of these two modalities, we introduce a dual-branch network. In the depth image branch, we input $I_{\mathcal{P}}$ and utilize the ResNet [12] to extract multi-hierarchical feature maps $\{F_D^l\}_{l=1}^L$ by hierarchically downsampling, where $L$ is the number of layers. In the point cloud branch, we treat each point cloud as a graph and employ the dynamic graph and edge convolution [36] to capture local details. Different from the depth image branch, we utilize intra-level (Figure 3) and inter-level (Figure 2) dense connections [16, 37] to organize point features at different hierarchies for extracting point cloud feature maps $\{F_{\mathcal{P}}^l\}_{l=1}^L$, instead of using residual connections [12]. Additionally, we refrain from downsampling the point cloud, as it is already sparse. To effectively fuse the aforementioned cross-modal features, we introduce the MVP block, which fuses cross-modal features at different hierarchical levels, as shown in Figure 2. There are two reasons why we use two different strategies to process depth images and point clouds, respectively: (i) The depth images with a resolution of $128 \times 128$ of a sparse point cloud with 2048 points are extremely sparse with at most only 12.5% of the pixels having an actual meaning. As an auxiliary branch, the depth image branch should be made as lightweight as possible. Therefore, we employ ResNet and hierarchically downsampling in this branch to expand the receptive field. (ii) Graph convolution [19, 28, 36] has been demonstrated to have brilliant performance in processing irregular point clouds. Furthermore, dense connections can more effectively utilize shallow features, particularly when cross-modal features are fused.

By using simple 2D convolutions on the depth image, capturing geometric detail information becomes straightforward. However, capturing such detail information directly in 3D space is challenging due to the sparsity, irregularity and non-uniformity present in the point cloud. Furthermore, the multi-hierarchical representations of depth images comprehensively provide multi-scale geometric details, ranging from coarse to fine.

## 3.3 Cross-Modal Feature Fusion

To effectively fuse the cross-modal features mentioned above, we propose the MVP block, as depicted in Figure 4. Given the feature maps $F_D = \{f_i\}_{i=1}^{N_v}$ of multi-view depth images and the point features $F_{\mathcal{P}}$ at the same hierarchical level, our approach aims to align these cross-modal features within the feature space. Subsequently,

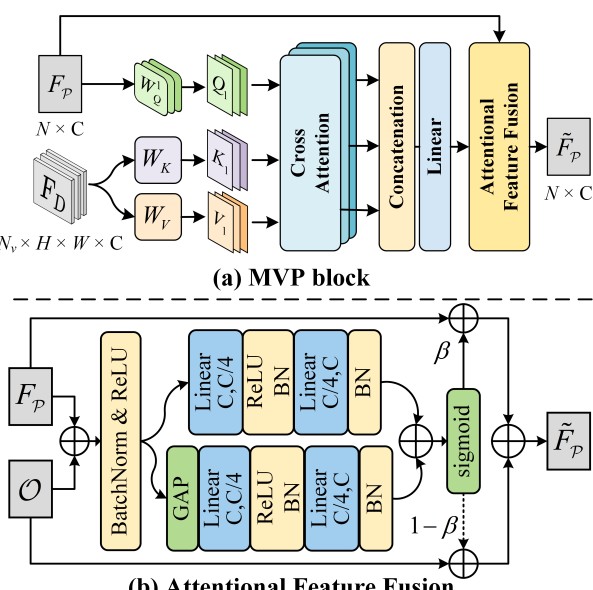

**(a) MVP block**

**(b) Attentional Feature Fusion**

**Figure 4: Illustration of (a) Multi-View Depth Image to Point Feature Fusion (MVP) and (b) Attentional Feature Fusion.**

we gather relevant pixel features in this aligned feature space. Finally, these features are projected back into the point feature space. The point features are first projected to queries by multiple parameter matrices $\{W_Q^i\}_{i=1}^{N_v}$. Then, the features of multi-view depth images are transformed into keys and values by parameter matrices $W_K$ and $W_V$, respectively. This process is formulated as:

$$Q_i = F_{\mathcal{P}} W_Q^i, \ K_i = f_i W_K, \ V_i = f_i W_V, \ i = 1, ..., N_v, \quad (1)$$

where $i$ is the index of viewpoints. Then, the cross-attention [34] is calculated between $Q_i$ and $K_i$ to generate aggregated depth image features $O_i$:

$$O_i = \mathrm{softmax}(Q_i K_i^{\mathsf{T}} / \sqrt{d_k}) V_i, i = 1, ..., N_v. \quad (2)$$

Since the calculation of attention mentioned above is conducted in the aligned feature space, an inverse projection is necessary for further feature fusion. This process can be expressed as:

$$O = \mathrm{Concat}(O_1, ..., O_{N_v}) W + b, \quad (3)$$

where $\mathrm{Concat}(\cdot)$ denotes the concatenation operation, and $W$ and $b$ are the parameters of a Linear Layer. Note that until now, we have only obtained the aggregated features $O$ of depth images and have not yet fused it into $F_{\mathcal{P}}$. A straightforward approach is to add them together, followed by a Feed Forward Network [34]. However, we find that this approach does not fuse them harmoniously. In practice, we modify the attentional feature fusion (AFF) [5] to fuse them:

$$\tilde{F}_{\mathcal{P}} = \beta \odot F_{\mathcal{P}} + (1 - \beta) \odot O, \ \beta = \mathrm{sigmoid}(\mathcal{A}(F_{\mathcal{P}}, O)), \quad (4)$$

where $\mathcal{A}$ is a sub-network that starts with an additional Batch-Norm Layer [17] to eliminate differences in data distribution between different modalities to predict $\beta$, and $\odot$ denotes element-wise multiplication.

## 3.4 Upsampling Tail

The upsampling tail takes the outcome of the cross-modal feature extraction module as input and expands it to achieve upsampling. We utilize the feature expansion module and the multi-scale spatial refinement module proposed in SSPU-Net [35] as our upsampling tail to generate upsampled point clouds in a coarse-to-fine manner, as shown in Figure 2. We don't use its frequency-aware attention mechanism to show that our approach's advantage comes from our contribution.

## 3.5 Detail Estimation and Distillation

We introduce a distilling structure to make full use of detail information from dense point clouds, achieving the generation of finer geometric details in upsampled point clouds. As illustrated in Figure 2, this distilling structure contains two training stages.

*3.5.1 Stage I: Teacher Network.* We concatenate the multi-view depth images of a sparse point cloud and the ground truth from the same viewpoint together along the channel dimension and input them to the depth image branch of the teacher network. Utilizing $I_{\hat{Q}}$, the teacher network can accurately reconstruct the geometric details and form multi-hierarchical detail representations $\{\tilde{F}_D^l = \{\tilde{f}_i^l\}_{i=1}^{N_v}\}_{l=1}^L$ of a dense point cloud.

*3.5.2 Stage II: Student Network.* The student network only takes a sparse point cloud $\mathcal{P}$ along with its multi-view depth images $I_\mathcal{P}$ as input. During its training, the parameters of the well-trained teacher network are frozen, and the teacher network only performs inference to produce $\{\tilde{F}_D^l\}_{l=1}^L$. To promote the student network to better learn the knowledge from the teacher network, we investigate two types of knowledge distillation methods: feature-based distillation [32] and response-based distillation [14].

For the feature-based distillation, the student network aims to mimic the multi-hierarchical detail representations $\{\tilde{F}_D^l\}_{l=1}^L$:

$$\mathcal{L}_{mimic} = \frac{1}{LN_v} \sum_{l=1}^L \sum_{i=1}^{N_v} \|\bar{f}_i^l - \tilde{f}_i^l\|_2^2, \tag{5}$$

where $\{\bar{F}_D^l = \{\bar{f}_i^l\}_{i=1}^{N_v}\}_{l=1}^L$ is the multi-hierarchical detail representations predicted by the student network. We also employ the spatial attention map-based distillation [41] to predict the spatial attention map of the $\{\tilde{F}_D^i\}_{i=1}^L$, which enables the student network to predict more meaningful pixels. The spatial attention map-based distillation is as follows:

$$\mathcal{L}_{AT} = \frac{1}{LN_v} \sum_{l=1}^L \sum_{i=1}^{N_v} \|\frac{\mathcal{G}(\bar{f}_i^l)}{\|\mathcal{G}(\bar{f}_i^l)\|_2} - \frac{\mathcal{G}(\tilde{f}_i^l)}{\|\mathcal{G}(\tilde{f}_i^l)\|_2}\|_2, \tag{6}$$

where $\mathcal{G}(\cdot)$ is an operator [41] to calculate the one-dimensional spatial attention map.

For the response-based distillation, the student network learns to predict the upsampled point clouds generated by the teacher network. This can be summarized as:

$$\mathcal{L}_{res} = \frac{1}{rN} \|Q_S^{'} - Q_\mathcal{T}^{'}\|_2^2 + \frac{1}{rN} \|Q_S - Q_\mathcal{T}\|_2^2, \tag{7}$$

where $Q_S^{'}$ and $Q_S$ are the coarse upsampled point cloud and fine upsampled point cloud generated by the student network respectively, and $Q_\mathcal{T}^{'}$ and $Q_\mathcal{T}$ are the results predicted by the teacher

network. Hence, the distillation loss of the student network is as follows:

$$\mathcal{L}_{distill} = \lambda_{mimic}\mathcal{L}_{mimic} + \lambda_{AT}\mathcal{L}_{AT} + \lambda_{res}\mathcal{L}_{res}. \tag{8}$$

The distillation constraints guide the student network in absorbing the knowledge necessary to reconstruct geometric details in dense point clouds. This knowledge is distilled from the teacher network and covers both feature and response perspectives. This approach helps the student network to reconstruct finer geometric details.

## 3.6 Training Objectives

The training objectives of the teacher network are only employed to ensure the quality of upsampled point clouds. We utilize the Chamfer Distance (CD), a prevalent choice in recent studies [6, 35], as one part of the reconstruction loss for the teacher network. We also employ the depth image matching loss [39] to measure the reconstruction error. Thus the training objectives of the teacher network can be summarized as follows:

$$\mathcal{L}_\mathcal{T} = \mathcal{L}_{rec} = \alpha(\mathcal{L}_{CD}(Q_\mathcal{T}, \hat{Q}) + \|\Phi(Q_\mathcal{T}) - \Phi(\hat{Q})\|_1)$$
$$+ (1 - \alpha)(\mathcal{L}_{CD}(Q_\mathcal{T}^{'}, \hat{Q}) + \|\Phi(Q_\mathcal{T}^{'}) - \Phi(\hat{Q})\|_1), \tag{9}$$

where $\alpha$ is a hyper-parameter to balance the relative importance of each term, and $\Phi(\cdot)$ is the differentiable point renderer [39]. We set $\alpha = 0.25$ for the PU-GAN dataset [21] and $\alpha = 0.5$ for the PUGeo-Net dataset [29]. The training objectives of the student network comprise the reconstruction term $\mathcal{L}_{rec}$ for its upsampled results, and the knowledge distillation term $\mathcal{L}_{distill}$:

$$\mathcal{L}_S = \mathcal{L}_{rec} + \lambda_{distill}\mathcal{L}_{distill}, \tag{10}$$

where $\lambda_{distill}$ is a hyper-parameter. We set $\lambda_{distill} = 0.15$ for both PU-GAN [21] and PUGeo-Net [29] dataset.

## 4 Experimental Results

## 4.1 Experimental Settings

*4.1.1 Datasets.* We conduct quantitative and qualitative experiments on two synthetic datasets: the PU-GAN dataset [21] and the PUGeo-Net dataset [29]. The latter contains more complex geometry and high-frequency details, compared to the former. We adhere to the same experimental settings as SSPU-Net [35], utilizing Poisson disk sampling [4] to generate patches for training and point clouds for testing. We also conduct qualitative experiments on the real-world ScanObjectNN [33] and KITTI [9] datasets.

*4.1.2 Evaluation Metrics.* Following previous works [13, 21, 28, 35], we adopt four evaluation metrics: (i) Density-aware Chamfer Distance (DCD) [38], (ii) Chamfer Distance (CD), (iii) Hausdorff Distance (HD), and (iv) Point-to-Surface (P2F) Distance. P2F metric include the mean (P2FM) and standard deviation (P2FS). For all metrics, the smaller the metric, the better the performance. The units of CD, HD, and P2F metrics are $10^{-3}$.

*4.1.3 Methods for Comparison.* We compare our method with a series of state-of-the-art methods, including PU-Net [40], MPU [37], PU-GAN [21], Dis-PU [22], PU-GCN [28], PUGeo-Net [29], MAFU [30], PUFA-GAN [23], PU-Flow [24], PUCRN [6], Grad-PU [13] and SSPU-Net [35].

**Table 1: Quantitative results on the PU-GAN dataset. The best and the second-best results are emphasized in bold and underlined respectively.**

| Ratio | 2× Upsampling | | | | | 4× Upsampling | | | | | 8× Upsampling | | | | |
|---|---|---|---|---|---|---|---|---|---|---|---|---|---|---|---|
| Method | DCD | CD | HD | P2FM | P2FS | DCD | CD | HD | P2FM | P2FS | DCD | CD | HD | P2FM | P2FS |
| PU-Net [40] | 0.355 | 0.553 | 3.610 | 3.577 | 4.875 | 0.355 | 0.398 | 3.325 | 4.126 | 4.978 | 0.403 | 0.411 | 5.643 | 4.987 | 5.724 |
| MPU [37] | 0.261 | 0.350 | 3.188 | 2.711 | 3.936 | 0.262 | 0.254 | 2.988 | 2.629 | 3.548 | 0.278 | 0.187 | 5.841 | 2.635 | 3.789 |
| PU-GAN [21] | 0.279 | 0.433 | 14.87 | 3.874 | 7.184 | 0.226 | 0.221 | 2.718 | 2.366 | 3.241 | 0.268 | 0.201 | 7.015 | 2.891 | 4.505 |
| Dis-PU [22] | 0.219 | 0.304 | 2.944 | 2.121 | 3.292 | 0.214 | 0.208 | 2.744 | 2.032 | 3.112 | 0.238 | 0.162 | 5.617 | 1.941 | 5.126 |
| PU-GCN [28] | 0.249 | 0.347 | 2.777 | 2.607 | 3.536 | 0.244 | 0.227 | 2.497 | 2.476 | 3.247 | 0.256 | 0.157 | 3.328 | 2.276 | 3.095 |
| PUGeo-Net [29] | 0.252 | 0.343 | 2.693 | 2.141 | 3.530 | 0.246 | 0.233 | 2.510 | 2.063 | 3.180 | 0.269 | 0.184 | 3.287 | 1.986 | 3.023 |
| MAFU [30] | 0.248 | 0.344 | 2.892 | 2.427 | 3.551 | 0.251 | 0.250 | 3.003 | 2.135 | 3.460 | 0.271 | 0.200 | 3.465 | 2.190 | 3.511 |
| PUFA-GAN [23] | 0.254 | 0.414 | 19.05 | 3.412 | 7.891 | 0.221 | 0.203 | 5.143 | 1.951 | 3.035 | 0.253 | 0.175 | 4.205 | 2.345 | 3.447 |
| PU-Flow [24] | 0.260 | 0.360 | 2.711 | 2.375 | 4.057 | 0.268 | 0.252 | 2.128 | 1.997 | 3.217 | 0.276 | 0.187 | 3.132 | 1.950 | 3.055 |
| PUCRN [6] | - | - | - | - | - | 0.222 | 0.223 | 2.673 | 1.880 | 3.166 | - | - | - | - | - |
| Grad-PU [13] | 0.298 | 0.413 | 2.632 | 1.957 | 3.687 | 0.232 | 0.210 | _1.940_ | 1.928 | 3.468 | 0.272 | 0.172 | **1.836** | 2.167 | 3.359 |
| SSPU-Net [35] | _0.198_ | _0.273_ | _2.603_ | **1.681** | _2.870_ | _0.198_ | _0.183_ | 2.001 | **1.426** | _2.274_ | _0.217_ | _0.132_ | 3.520 | _1.458_ | _2.606_ |
| Ours (Teacher) | 0.174 | 0.242 | 2.276 | 1.765 | 2.449 | 0.176 | 0.160 | 1.490 | 1.610 | 2.204 | 0.196 | 0.113 | 1.611 | 1.283 | 1.966 |
| Ours (Student) | **0.185** | **0.266** | **2.297** | _1.713_ | **2.275** | **0.186** | **0.172** | **1.919** | _1.609_ | **2.206** | **0.201** | **0.120** | _2.928_ | **1.283** | **2.268** |

### 4.1.4 Implementation Details.

We utilize three orthogonal viewpoints to generate multi-view depth images with a resolution of $128 \times 128$. The hyper-parameters $\lambda_{mimic}$, $\lambda_{AT}$ and $\lambda_{res}$ are set to 0.1, 1 and 500, respectively. During training, we set $N = 256$ for each sparse patch, which is randomly downsampled from the dense patch. We train both the teacher network and the student network with a batch size of 32. The former is trained for 200 epochs, while the latter is trained for 240 epochs. The Adam algorithm [18] is adopted to optimize our network. The initial learning rate is set to 0.001 with a decay rate of 0.7.

**Table 2: 4× quantitative results on the PUGeo-Net dataset. We emphasize the best and second-best results by bolding and underlining them, respectively.**

| Ratio | 4× Upsampling | | | | |
|---|---|---|---|---|---|
| Method | DCD | CD | HD | P2FM | P2FS |
| PU-Net [40] | 0.393 | 0.285 | 5.079 | 2.470 | 2.522 |
| MPU [37] | 0.320 | 0.186 | 2.991 | 1.307 | 1.793 |
| PU-GAN [21] | 0.357 | 0.254 | 10.89 | 1.855 | 2.460 |
| Dis-PU [22] | 0.293 | 0.166 | 3.333 | 1.186 | 1.830 |
| PU-GCN [28] | 0.330 | 0.188 | 3.920 | 1.694 | 2.064 |
| PUGeo-Net [29] | 0.314 | 0.179 | 3.255 | 1.234 | 1.839 |
| MAFU [30] | 0.322 | 0.175 | 4.779 | 1.511 | 2.182 |
| PUFA-GAN [23] | 0.308 | 0.179 | 7.172 | 1.372 | 2.086 |
| PU-Flow [24] | 0.301 | 0.150 | 1.631 | 1.024 | 1.294 |
| PUCRN [6] | 0.250 | 0.122 | 3.236 | 1.009 | 1.674 |
| Grad-PU [13] | _0.211_ | _0.088_ | **0.948** | _0.682_ | 1.153 |
| SSPU-Net [35] | 0.228 | 0.102 | 1.595 | **0.675** | **1.033** |
| Ours (Teacher) | 0.184 | 0.076 | 1.112 | 0.644 | 0.736 |
| Ours (Student) | **0.198** | **0.084** | _1.586_ | 0.712 | _1.092_ |

## 4.2 Results on Synthetic Dataset

### 4.2.1 Results on the PU-GAN dataset.

Table 1 presents the quantitative comparison results on the PU-GAN dataset. Our method outperforms previous approaches in terms of DCD and CD metrics, indicating that our method achieves minimal reconstruction errors at different upsampling ratios. Since DCD can measure the density difference between two point clouds, our upsampled results exhibit the closest density distribution to the ground truth, suggesting that our results are more uniform. For the HD metric, our

method outperforms all other methods at 2× and 4× upsampling ratios and achieves the second-best performance at 8×. For the P2F metric, our method exhibits the most competitive performance with SSPU-Net [35] on P2FM and the best results on P2FS. This demonstrates that our results are stably distributed on the surface of the underlying object. As depicted in the upper section of Figure 5, previous methods tend to generate more noise and outlier points when handling complex geometry. In contrast, our method can accurately reconstruct complex geometric details with minimal noise and outliers.

### 4.2.2 Results on the PUGeo-Net dataset.

Table 2 shows the 4× quantitative results on the PUGeo-Net dataset. Even when faced with more complex geometric details, our method consistently achieves the best performance in terms of DCD and CD metrics. Furthermore, for the P2F metric, our method also demonstrates comparable results to recent state-of-the-art methods, namely SSPU-Net [35] and Grad-PU [13]. However, for the HD metric, Grad-PU [13] achieves better performance, and we believe this phenomenon arises from the differences in the upsampling tail. Grad-PU [13] refines the positions of interpolated points by a point-to-point (p2p) iterative optimization process. This process enables it to achieve the smallest maximum p2p error, resulting in the lowest HD metric. However, it lacks overall consideration and thus cannot surpass us in DCD and CD metrics. This is evidenced in the lower section of Figure 5, where our method can produce the most realistic results.

## 4.3 Results on Real-Scanned Dataset

We also conduct experiments on two real-world datasets, the ScanObjectNN dataset [33] and the KITTI dataset [9], as shown in Figure 6 and Figure 7, respectively. Since there is no ground truth available, we only perform qualitative comparisons. Figure 6 illustrates the upsampled results (bottom row) and surface reconstruction results (top row), where two consecutive 4× upsamplings are conducted using models trained on the PU-GAN 4× dataset. As shown in the first column, the input point cloud is extremely sparse and non-uniformly distributed. There are noticeable gaps in the upsampled results generated by some methods, such as PUCRN [6], PU-Flow[24] and SSPU-Net [35]. Grad-PU [13] fails to generate

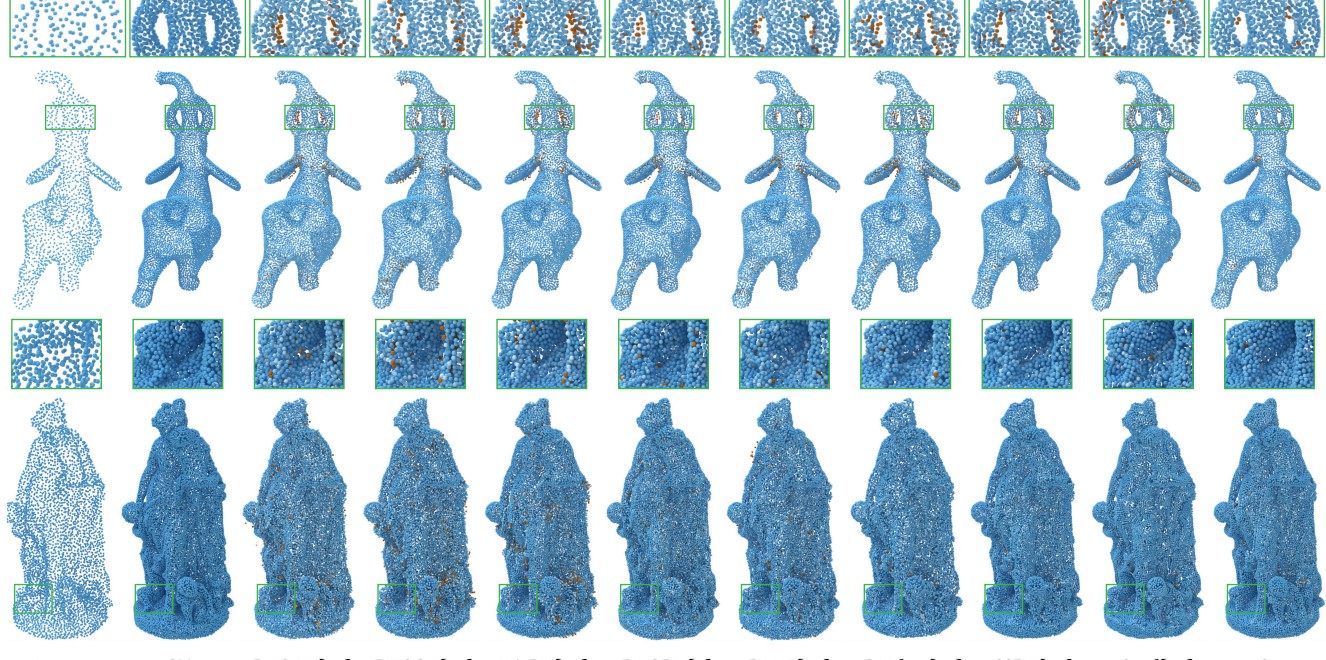

**Figure 5:** 4× qualitative comparison on synthetic datasets, where the upper section is the results on the PU-GAN dataset and the lower section is the results on the PUGeo-Net dataset. We visualize the P2F metrics of each point. If a point has a small P2F, it's shown in blue, otherwise, it's shown in red. Ours denotes the result of the student network.

plausible results. The surface reconstruction results of PU-GAN [21] and Dis-PU [22] reveal holes. While our method can reconstruct dense and uniformly distributed upsampled results, as well as achieve smooth surface reconstruction results. Figure 7 shows our method can accurately preserve the original structure (orange box) and fill in the gaps caused by occlusion (red box).

**Table 3: Robustness test with different Gaussian noise levels on the PU-GAN 4× dataset. We use DCD as the metric.**

| Method | 0.00% | 0.30% | 0.50% | 0.80% | 1.00% | 2.00% |
|---|---|---|---|---|---|---|
| PU-Net [40] | 0.355 | 0.380 | 0.378 | 0.418 | 0.461 | 0.539 |
| MPU [37] | 0.261 | 0.281 | 0.299 | 0.343 | 0.381 | 0.469 |
| PU-GAN [21] | 0.226 | 0.255 | 0.275 | 0.330 | 0.366 | 0.464 |
| Dis-PU [22] | 0.214 | 0.312 | 0.323 | 0.365 | 0.391 | 0.486 |
| PU-GCN [28] | 0.244 | 0.279 | 0.299 | 0.340 | 0.377 | 0.467 |
| PUGeo-Net [29] | 0.246 | 0.275 | 0.294 | 0.343 | 0.375 | 0.473 |
| MAFU [30] | 0.248 | 0.271 | 0.290 | 0.332 | 0.371 | 0.462 |
| PUFA-GAN [23] | 0.254 | 0.262 | 0.274 | 0.319 | 0.358 | 0.468 |
| PU-Flow [24] | 0.268 | 0.279 | 0.306 | 0.358 | 0.401 | 0.485 |
| PUCRN [6] | 0.222 | 0.253 | 0.272 | 0.318 | 0.360 | 0.457 |
| Grad-PU [13] | 0.232 | 0.255 | 0.298 | 0.358 | 0.406 | 0.502 |
| SSPU-Net [35] | 0.198 | 0.224 | 0.254 | 0.310 | 0.356 | 0.462 |
| Ours (Student) | **0.186** | **0.212** | **0.238** | **0.287** | **0.332** | **0.444** |

## 4.4 Robustness to Noise

Since point clouds usually contain noise and outliers in real scenarios, we conduct experiments to assess the noise robustness of our proposed method. Specifically, we add Gaussian noise at different levels from 0 to 2.0% to the PU-GAN 4× testing dataset. Table 3 demonstrates that our method achieves the best performance at different noise levels. DCD is used as the metric instead of CD because the upsampling results of Grad-PU tend to exhibit an aggregation

of points around the original point locations under higher noise levels. This density imbalance can even result in a lower CD, as explained in DCD [38].

**Table 4: Comparison of model complexity on the PU-GAN 4× dataset conducted on an RTX 3090 GPU.**

| Method | CD ($10^{-3}$) | Training (h) | Inference (s/sample) | Param. (Kb) |
|---|---|---|---|---|
| PU-Net [40] | 0.398 | 4.7 | 0.44 | 814 |
| MPU [37] | 0.254 | 3.0 | 0.20 | 76 |
| PU-GAN [21] | 0.221 | 12.0 | 0.20 | 542 |
| Dis-PU [22] | 0.208 | 15.3 | 0.52 | 1047 |
| PU-GCN [28] | 0.227 | 0.5 | 0.20 | 76 |
| PUGeo-Net [29] | 0.233 | 10.5 | 0.43 | 2126 |
| MAFU [30] | 0.250 | 12.0 | 0.60 | 260 |
| PUFA-GAN [23] | 0.203 | 31.0 | 0.86 | 4642 |
| PU-Flow [24] | 0.252 | 11.5 | 0.32 | 806 |
| PUCRN [6] | 0.223 | 2.3 | 0.27 | 847 |
| SSPU-Net [35] | 0.183 | 13.5 | 0.53 | 533 |
| Grad-PU [13] | 0.210 | 5.7 | 0.19 | 67 |
| Ours (Teacher) | 0.160 | 11.5 | 0.90 | 1787.77 |
| Ours (Student) | 0.172 | 14.9 | 0.78 | 1787.62 |

## 4.5 Model Complexity Analysis

The results of the model complexity analysis are shown in Table 4. Since we only use the student network in practice, the number of parameters utilized in the inference phase of PUFA-GAN [23] is approximately 2.5 times that of our method. Additionally, our method is comparable to Dis-PU [22]. The inference time of our method is slightly longer than previous methods, such as MAFU [30], but less than PUFA-GAN. This is because our method needs to render multi-view depth images of each point cloud and extract

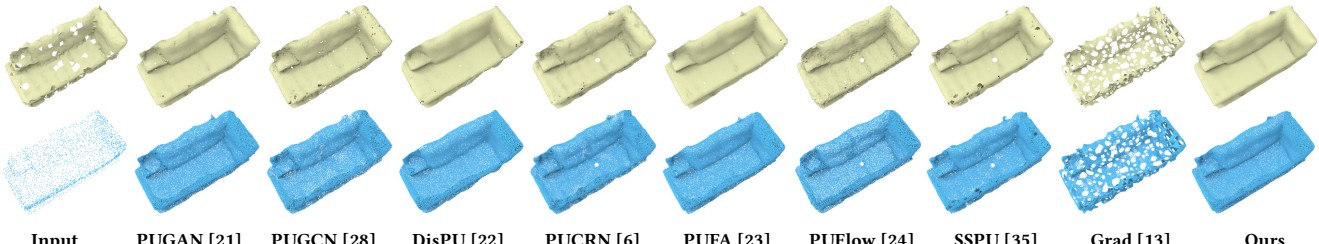

**Figure 6:** 16× **upsampled results on the ScanObjectNN dataset [33]. The first row and the second row showcase the meshes reconstructed by [2] and the upsampled point clouds, respectively. Ours denotes the result of the student network.**

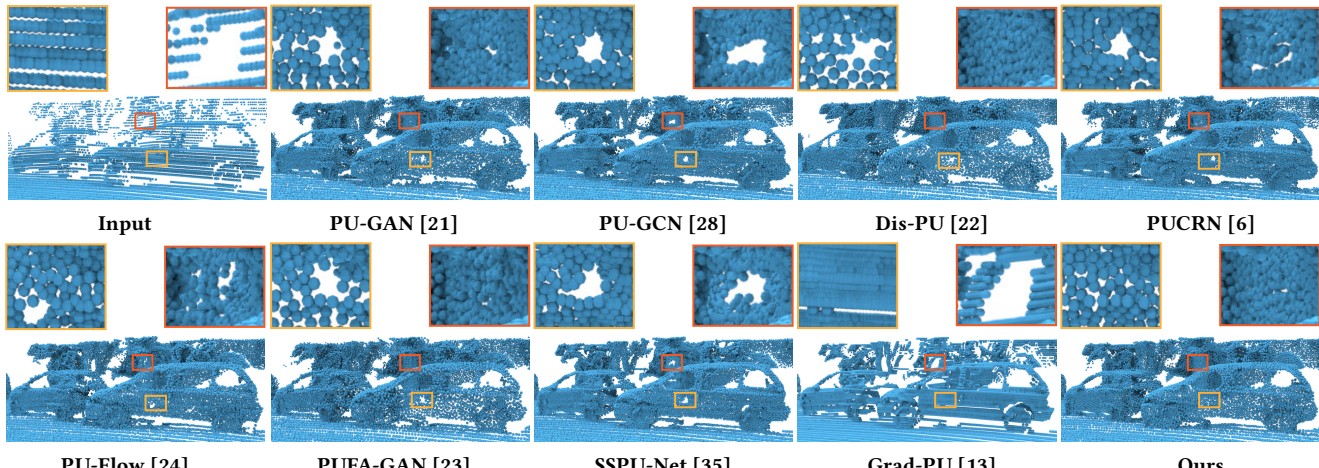

**Figure 7: Visual comparison on the KITTI dataset [9]. Our method can accurately preserve the original structure (orange box) and fill in the gaps caused by occlusion (red box). Ours denotes the result of the student network.**

**Table 5: Ablation studies of the cross-modal feature extraction and fusion, and the detail estimation distillation.**

| Model | $\mathcal{L}_{mimic}$ | $\mathcal{L}_{AT}$ | $\mathcal{L}_{res}$ | CD $(10^{-3})$ | HD $(10^{-3})$ |
|---|---|---|---|---|---|
| A1 | | | | 0.177 | 2.095 |
| A2 | | ✓ | ✓ | 0.174 | 2.168 |
| A3 | ✓ | | ✓ | 0.175 | 2.041 |
| A4 | | | ✓ | 0.176 | 2.003 |
| A5 | ✓ | ✓ | | 0.174 | 1.926 |

| Model | AFF | multi-$W_Q$ | $\Phi$ | CD $(10^{-3})$ | HD $(10^{-3})$ |
|---|---|---|---|---|---|
| B1 | | | ✓ | 0.186 | 2.267 |
| B2 | | ✓ | ✓ | 0.189 | 2.130 |
| B3 | ✓ | | ✓ | 0.179 | 2.103 |
| B4 | ✓ | ✓ | | 0.179 | 2.111 |
| Full | - | - | - | **0.172** | **1.919** |

their features. The total training time of our method is about 26.4 hours, including the training time of the teacher network and the student network.

### 4.6   Ablation Study

We initially carry out ablation studies focusing on the detail estimation and distillation, as illustrated in the upper section of Table 5. Based on models A2, A3 and A4, we conclude that the depth image branch of the teacher network effectively captures the fine geometric details in dense point clouds. This inference is drawn

from the observation that removing $\mathcal{L}_{mimic}$ or $\mathcal{L}_{AT}$ results in suboptimal performance of the student network. From model A5, the response-based knowledge distillation further improves the upsampling performance. We then conduct ablation studies on the cross-modal feature extraction without the guidance of the teacher network, as illustrated in the lower section of Table 5. Compared to model A1, model B1, which utilizes only 3D coordinates for upsampling, demonstrates that leveraging cross-modal information can significantly improve point cloud upsampling. Model B2, which fuses $O$ and $F_{\mathcal{P}}$ by direct addition, achieves worse results than the uni-modal model B1, highlighting the importance of a suitable feature fusion method for cross-modal features. Models B3 and B4 confirm the effectiveness of employing multiple $W_Q$ in the MVP block and the depth image matching loss, respectively.

### 5   Conclusion

We propose a novel multi-view depth image guided cross-modal distillation network for point cloud upsampling. Specifically, we first introduce the cross-modal feature extraction module integrated with the MVP blocks. The former consists of two branches to extract both depth image features and point features. The latter fuses these cross-modal features in a fine-grained and hierarchical manner. Additionally, we introduce the detail estimation and distillation structure to produce more realistic geometric details. Extensive experiments demonstrate that MVP-Net achieves state-of-the-art performance.

## Acknowledgments

This work was supported by the Natural Science Foundation of China (62272016, 62372018), Beijing Natural Science Foundation (4232017), and the Opening Project of Beijing Key Laboratory of Internet Culture and Digital Dissemination Research.

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
