# OpenReview forum: "MVP-Net: Multi-View Depth Image Guided Cross-Modal Distillation Network for Point Cloud Upsampling"
_acmmm.org/ACMMM/2024/Conference — MM2024 Poster_

### Official Review · Reviewer_DxAu · 2024-05-06

**Rating:** 4
**Confidence:** 2

**Summary:**

The paper proposes a novel cross-modal approach named MVP-Net for point cloud upsampling, utilizing multi-view depth images to guide the upsampling process. It addresses two main challenges encountered by current upsampling methods: insufficient uni-modal representations of sparse point clouds and inaccurate estimation of geometric details in dense point clouds.

**Strengths:**

The strengths of this paper are as follows:
1. This paper introduces a novel method for cross-modal point cloud upsampling.
2. To accommodate this cross-modal method, the paper proposes new modules for feature extraction and feature fusion.
3. The paper conducted numerous experiments to validate the entire method.

**Limitations:**

The limitations of this paper are as follows:
1. Some aspects of the paper lack novelty. For example, the cross-modal feature extraction module mentioned in the contributions section seems to essentially apply DGCNN[1] to point clouds and ResNet[2] to depth images, respectively. I would like the authors to elaborate further on the novelty.
2. Some descriptions in the paper are confusing, such as in the ablation experiments, where the authors do not explicitly explain the abbreviations in the table.
3. The authors do not provide a thorough explanation for the modifications of the feature fusion module.

Further comments:
1. Whether adding the dense point cloud‘s multi-view depth images to the teacher network is a fair approach compared to other baselines? Does it lead to overfitting?

Reference:
[1]Yue Wang, Yongbin Sun, Ziwei Liu, Sanjay E. Sarma, Michael M. Bronstein, and Justin M. Solomon. 2019. Dynamic Graph CNN for Learning on Point Clouds. ACM Trans. Graph. 38, 5 (2019), 146:1–146:12
[2]Kaiming He, Xiangyu Zhang, Shaoqing Ren, and Jian Sun. 2016. Deep Residual Learning for Image Recognition. In 2016 IEEE Conference on Computer Vision and Pattern Recognition, CVPR 2016, Las Vegas, NV, USA, June 27-30, 2016. IEEE Computer Society, 770–778

**Suitability:**

3

---

### Official Review · Reviewer_zLcd · 2024-05-23

**Rating:** 3
**Confidence:** 2

**Summary:**

This paper introduces MVP-Net, a novel approach for point cloud upsampling that utilizes multi-view depth images to guide the upsampling process. The network addresses challenges related to sparse point cloud representations and inaccurate geometric detail estimation in dense point clouds. Experimental results demonstrate that MVP-Net outperforms existing point cloud upsampling methods.

**Strengths:**

1. The authors propose to use multi-view depth images as guidance to point cloud upsampling and design a cross-modal distillation network named MVP-Net for point cloud upsampling.
2. A distilling structure and cross-model feature fusion block are introduced to produce finer geometric details, the authors claim that this MVP-Net achieves the state-of-the-art performance.

**Limitations:**

1. The authors have only utilized depth maps from three orthogonal views. Can incorporating depth maps from additional views enhance the model's accuracy? Are there any relevant experiments conducted to investigate this?
2. This paper mentions that the concept of incorporating depth maps is inspired by SVDFormer. However, the depth map is view-dependent and lacks multi-view consistency. Additionally, SVDFormer utilizes camera poses as positional embeddings for differentiation. Is it appropriate to directly apply depth map features to enhance point cloud features in this paper?
3. The experimental results presented in the paper, particularly on the PU-GAN dataset in Table 1 and the PUGeo-Net dataset in Table 2, do not show a significant improvement in accuracy.
4. In Table 4, the MVP-Net proposed in this paper exhibits long training and inference times, along with a large number of parameters. Have the authors considered strategies to enhance efficiency in light of these observations?

**Suitability:**

3

---

### Official Review · Reviewer_gWFG · 2024-05-23

**Rating:** 2
**Confidence:** 4

**Summary:**

This manuscript presents a point cloud upsampling method by combining multi-view depth images rendering and knowledge distillation technique. The innovation is limited. Firstly, the depth images are rendered from point clouds, so it is doubtful whether they can provide additional information. Secondly, the teacher-student paradigm is a widely-applied framework, which does not need any modification to extend to point cloud upsampling in this paper.

**Strengths:**

1.This paper is well-written, and the main idea of this work is clearly presented.

2.The introduction provides a thorough background and the methodology is presented in a clear and accessible manner.

**Limitations:**

1.The motivation is unclear. The authors claimed that previous methods suffer from two shortcomings: (i) insufficient uni-modal representations of sparse point clouds, and (ii) inaccurate estimation of geometric details in dense point clouds, resulting in suboptimal upsampling results.
Correspondingly, (i) the depth images are rendered from sparse point clouds. How can they provide sufficient representations for upsampling? (ii) The key of point cloud upsampling is to estimate the geometric details of dense point clouds from sparse ones. How can we claim that previous works failed to extract the geometric details of dense point clouds?

2.The teacher-student paradigm is a widely-applied framework. Does this paper do anything to make it more suitable for a point cloud upsampling task?

3.More training programs need to be studied. Refer to the [1] for more details.
[1] Wang X, Cui W, Xiong R, et al. FCNet: Learning Noise-Free Features for Point Cloud Denoising[J]. IEEE Transactions on Circuits and Systems for Video Technology, 2023.

**Suitability:**

3

---

### Official Review · Reviewer_tY3c · 2024-05-23

**Rating:** 4
**Confidence:** 3

**Summary:**

This paper introduces a method called MVP-Net, which addresses the challenges of point cloud upsampling by using multi-view depth images to guide the upsampling process. It aims to produce a dense and uniform point set from a sparse and irregular one, overcoming issues such as insufficient representations of sparse point clouds and inaccurate estimation of geometric details in dense point clouds. The proposed method consists of a cross-modal feature extraction module, a Multi-View Depth Image to Point Feature Fusion (MVP) block, and a multi-view depth image-guided detail estimation and distillation paradigm. The method has shown superior performance compared to state-of-the-art point cloud upsampling methods in both synthetic and real-world datasets.

**Strengths:**

1. The authors propose a novel network (MVP-Net) for upsampling point cloud data into a dense cloud. They introduce a cross-modal feature extraction module, a Multi-View Depth Image to Point Feature Fusion (MVP) block, and a distillation paradigm for point cloud upsampling.

2. The paper (MVP-Net) compares results with state-of-the-art methods to demonstrate the effectiveness of the proposed technique. Standard evaluation metrics such as density-aware chamfer distance and chamfer distance are utilized to make the comparison. Additionally, the complexity of MVP-Net is compared with other models. Also, they show the qualitative comparisons in the paper.

3. The paper clearly explains the modules of the proposed MVP-Net in achieving point cloud upsampling. It also details some limitations associated with MVP-Net.

**Limitations:**

1. The paper does not study different values for the hyper-parameters ($\lambda_{mimic}$, $\lambda_{AT}$, $\lambda_{res}$) in the distillation loss but sets fixed values. However, these hyper-parameters influence MVP-Net's performance.


2. Also, the authors do not study different values for $\alpha$ but set a value (0.25 for the PU-GAN dataset and 0.5 for the PUGeo-Net dataset) for the reconstruction term (Equation 9). Similarly, $\lambda_{distill}$ is not studied for different values in Equation 10. Experimental details on these above parameters are required to understand how the authors arrived at these hyper-parameters values.

**Suitability:**

3

---

### Meta-Review · Area_Chair_3qVj · 2024-07-01

**Recommendation:** Accept (Poster)
**Confidence:** 5

**Metareview:**

This paper initially got mixed reviews. After the rebuttal phase, the reviewer gWFG also upgraded his/her rating. Although he/she still had some concerns about the motivation, which should be easily addressed in the revised manuscript. By taking reviews into consideration, ACs recommend the acceptance of the paper.